# Allergic Dermatitis in Pêga Breed Donkeys (*Equus asinus*) Caused by *Culicoides* Bites in the Amazon Biome, Pará, Brazil

**DOI:** 10.3390/ani14091330

**Published:** 2024-04-29

**Authors:** José Diomedes Barbosa, Maria Hilma Soares Sodré, Camila Cordeiro Barbosa, Paulo Sérgio Chagas da Costa, Carlos Magno Chaves Oliveira, Tatiane Teles Albernaz Ferreira, José Alcides Sarmento da Silveira, Eryca Ceolin Lamego, Milena Carolina Paz, Rossela Damasceno Caldeira, Paulo César Magalhães Matos, Analiel Serruya, Felipe Masiero Salvarani, Natália da Silva e Silva Silveira

**Affiliations:** 1Institute of Veterinary Medicine, Federal University of Pará (UFPA), Castanhal 68740-970, PA, Brazil; diomedes@ufpa.br (J.D.B.); hilmasodre@gmail.com (M.H.S.S.); camilabarbosamedvet@gmail.com (C.C.B.); cmagno@ufpa.br (C.M.C.O.); tatyalbernaz@ufpa.br (T.T.A.F.); jalcides@ufpa.br (J.A.S.d.S.); analielserruya.as@gmail.com (A.S.); felipems@ufpa.br (F.M.S.); 2Antônio Leite College/UNIBTA (FAL), Castanhal 68742-000, PA, Brazil; paulocosta4621@gmail.com; 3Department of Veterinary Pathology, Federal University of Rio Grande do Sul (UFRGS), Porto Alegre 90040-060, PR, Brazil; erycalamego@gmail.com (E.C.L.); mileenapaz@hotmail.com (M.C.P.); 4Postgraduate Program in Biology of Infectious Agents and Parasites, PPGBAIP/UFPA, Belém 66075-110, PA, Brazil; rosselabio@gmail.com; 5Federal Institute of Amapá (IFAP), Agricultural Campus of Porto Grande, Porto Grande 68997-000, AP, Brazil; paulo.matos@ifap.edu.br

**Keywords:** donkey, itching, alopecia, crusts, skin lesions

## Abstract

**Simple Summary:**

This study aimed to describe the epidemiological, clinicopathological, and therapeutic aspects of allergic dermatitis caused by *Culicoides* spp. bites in donkeys. Clinical signs included restlessness and severe itching. Skin lesions were observed across various parts of the body and included areas of alopecia with crusts and serosanguinous exudates. Overall, 378 *Culicoides* insects were collected and *Culicoides ocumarensis* Ortiz was identified as the most common species. These findings suggest an association between allergic dermatitis and *Culicoides*. Additionally, a combination of copaiba oil and a multivitamin emulsion showed therapeutic potential. To date, this is the first study on allergic dermatitis in donkeys in Brazil.

**Abstract:**

An allergy to bites from *Culicoides* (Diptera: Ceratopogonidae) occurs because of a hypersensitivity reaction caused by the inoculation of insect salivary antigens during the bite, resulting in immune-mediated dermatitis. To the best of our knowledge, no previous studies have focused on allergic dermatitis in donkeys in Brazil. Therefore, this study aimed to describe the epidemiological, clinicopathological, and therapeutic aspects of allergic dermatitis in donkeys and to identify the insects involved in its epidemiology. This study reported the occurrence of dermatitis in 17 animals. The clinical signs were restlessness and severe itching. Skin lesions were found on the head, depigmented areas of the muzzle and cheeks, flanks, pelvic and thoracic limbs, and the scrotal sac. The lesions were characterized by areas of alopecia with crusts accompanied by serosanguineous exudates. Histologically, the lesions were characterized as moderate superficial dermatitis with irregular epidermal acanthosis and pronounced diffuse orthokeratotic hyperkeratosis. In total, 378 *Culicoides* specimens were collected, with *Culicoides ocumarensis* Ortiz being the most abundant species. The combined application of copaiba oil and a multivitamin emulsion exhibited potential for topical treatment of allergic dermatitis caused by insect bites in donkeys. Our study revealed an association between allergic dermatitis in donkeys and *Culicoides*.

## 1. Introduction

According to the Food and Agriculture Organization, the global population of donkeys is approximately 45 million [1], with approximately one million donkeys in Brazil, 90% of which are located in the northeast region [2].

In the state of Pará, the donkey population is established through the purchase of breeding animals from other regions of Brazil, with the aim of crossbreeding them with mares to produce mules, which play a crucial role in the daily management of cattle farms. Owing to their inherent traits such as strength, hardiness, resistance to hot climates, and food scarcity, donkeys play a pivotal role in transportation and traction, particularly on small farms and in subsistence agriculture [3].

In northeastern Brazil, the region with the largest donkey population, the most commonly diagnosed ailments include traumatic wounds, fractures, and colic. However, despite the importance of donkeys in the region, there are few scientific studies on diseases affecting these animals published in relevant national and international journals. Donkey keepers generally uphold the traditional belief that donkeys and mules are highly resilient animals requiring minimal health and management care [4].

Donkeys in the state of Pará in the Amazon Biome experience chronic and pruritic skin conditions during the peak rainy season. The conditions are characterized by multifocal erythema, papules, alopecia, and a thickened skin with crusts, which was diagnosed as allergic dermatitis after ruling out the most relevant differential diagnoses.

Seasonal allergic dermatitis, a skin disease associated with insect bites, is prevalent in horses, sheep, cattle, mules, and humans in tropical and subtropical regions. The skin areas around the eyes, ears, muzzle, lips, abdomen, perineum, and limbs are the most affected, with hair and wool easily detaching from the affected body regions. Additionally, weight loss, ocular discharge, and in some cases, keratitis with corneal opacity have been observed. These conditions can lead to mutilation, infection, and secondary myiasis owing to skin lesions [5,6,7,8,9,10,11].

Allergic dermatitis due to insect bites is the most common cutaneous allergic disease in horses and presents with multifactorial characteristics influenced by genetic and environmental components [12,13,14]. Differential diagnoses of allergic dermatitis include atopic dermatitis, urticaria, contact, and food hypersensitivities [15].

Allergic dermatitis occurs through a type I hypersensitivity reaction caused by the release of salivary proteins from the pathogen that act as antigens [13]. In the chronic phase, a type of intravenous (IV) delayed allergic response occurs, leading to tissue inflammation and damage. This results in fibrin deposition, which maintains swelling and sensitivity in the affected area [13,16]. *Culicoides* genus insects are the main agents involved; however, other insects such as *Stomoxys*, *Haematobia irritans*, and *Simulium* may also contribute to disease onset [17].

*Culicoides* are the most abundant hematophagous insects in the world and are vectors of various pathogens responsible for the transmission of over 50 viruses, including those affecting humans [18]. More than 40 species of *Culicoides* are associated with notifiable diseases, such as the Bluetongue virus and African horse sickness virus. In addition to viruses, *Culicoides* also transmit bacteria, nematodes, and protozoa and can cause allergic reactions in horses worldwide. Their impact on veterinary medicine and public health is significant, causing severe economic losses [18,19,20].

Previous studies have not focused on allergic dermatitis in donkeys in Brazil. Therefore, the aim of this study was to describe the epidemiological, clinicopathological, and therapeutic aspects of allergic dermatitis in donkeys and to identify the insects involved in its epidemiology.

## 2. Materials and Methods

### 2.1. Ethical Aspects

All animal experiments were approved and conducted in compliance with the experimental practices and standards developed by the rules issued by the National Council for Control of Animal Experimentation and were approved by the Ethic Committee on Animal Use of the Federal University of Para (CEUA/UFPA), protocol number CEUA 7719300323 (ID 002217).

### 2.2. Epidemiology and Clinical Examination

This study included observations conducted on 69 donkeys from 5 rural farms (identified as farm 1–5) located in the northeast of Pará State, in the Brazilian Amazon Biome. The study group included 17 donkeys showing clinical signs (animals 1–17) and 52 healthy ones. Data on age, sex, race, farm location, and type of management adopted on the farm were obtained during the farm visits. The integumentary system of animals with skin lesions was specifically examined using the protocol established by Feitosa et al. [21].

### 2.3. Insect Capture and Identification

The insects were captured in February 2023 during their blood meals on donkeys in the stables, and collections were carried out for each property using a Center for Disease Control light trap, which was positioned 190 cm above ground level. The traps were operated continuously from 5:00 p.m. to 7:00 a.m. the following day. After capture, the insects were placed in containers, frozen at −20 °C, and sent to the Medical Entomology Laboratory, part of the Arbovirology and Hemorrhagic Fevers Section at the Evandro Chagas Institute, Ananindeua, Pará, for processing, screening, and identification. The insects were screened under a stereoscopic microscope, separated into morphospecies, and mounted onto slides. The assembly was conducted using the phenol–balsam method described by Wirth and Marston [22]. The insects were identified based on the color of different parts of the female body, measurements and morphometric relationships of the head, wing, and spermatheca(s), and quantification of structures from various body parts [23]. Additionally, insects were identified according to methods described in previous studies [24,25,26,27,28].

### 2.4. Collection and Processing of Biological Samples from Donkeys

Blood samples were collected from the jugular vein of 15 sick and 5 healthy animals into sterile vacuum tubes with and without the anticoagulant ethylenediaminetetraacetic acid for blood count tests and biochemical analyses, respectively. Subsequently, the samples were chilled at 6–8 °C for up to 3 h and sent to the Clinical Pathology Laboratory of the University Veterinary Hospital at the Federal University of Pará.

Hemograms were obtained using standard techniques [29]. The aspartate aminotransferase, gamma-glutamyl transferase, urea, creatinine, and bilirubin levels in the serum were analyzed on an automated chemistry analyzer (Mindray BS-120) using commercial reagents (Bioclin^®^), following the manufacturer’s instructions.

Skin scrapings from the lesions were collected, stored in Falcon tubes, and sent to the laboratory for ectoparasite analysis.

A skin biopsy was performed at the interface of the intact and lesioned skin on the head region, following the IV application of 1% detomidine (0.03 mg/kg) and a local block with 2 mL of 2% lidocaine hydrochloride (40 mg). The collected material was preserved in buffered 10% formalin and sent to the Veterinary Pathology Department of the Federal University of Rio Grande do Sul for histopathological examination. The tissue samples were routinely processed, embedded in paraffin, cut to 5 µm sections, and stained with hematoxylin and eosin.

The samples from donkey 4 were histochemically tested using toluidine blue, Gram, and Warthin–Starry staining to reveal mast cells, coccoid/filamentous bacteria, intracellular bacteria, and spirochetes, respectively.

### 2.5. Statistical Analysis

Hematological and serum biochemical test results were deemed parametric after analysis using the “W” test (Shapiro–Wilk). Significant differences between the test results of healthy and sick animals were assessed using one-way analysis of variance.

Statistical analyses were conducted using BioEstat program version 5.3, with a significance level of *p* = 0.05.

## 3. Results

### 3.1. Epidemiology and Clinical Findings

Overall, 69 Pêga breed donkeys (*Equus asinus*) aged 1–8 years, comprising 14 males and 55 females, originating from the Santa Luzia do Pará (farm 1), Inhangapi (farms 2 and 5), Terra Alta (farm 3), and Santo Antônio do Tauá (farm 4) municipalities in Pará, were observed. During anamnesis, animal owners or caretakers reported that all adult animals exhibited persistent cutaneous manifestations for a period > 1 year, whereas young animals presented with these lesions for at least 6 months. Table 1 presents the number of animals per farm and the morbidity rate of allergic dermatitis.

All animals included were raised in an extensive system of *Urochloa humidicola* and *Panicum maximum* “Mombaça”. Water originated from streams and/or lakes located in pastures. On all farms, donkeys coexisted with horses and showed no signs of injury. However, on farm 4, goats and sheep showed similar skin symptoms, suggesting that allergic dermatitis may be associated with insect bites. During the period of the highest rainfall, all farms consistently experienced water accumulation (puddles) in pastures and around management facilities. Animal caretakers from farms 3 and 4 reported that the lesions healed when the affected donkeys were relocated to other places (neighboring farms) or confined in stables and treated using methods adopted by the owners, such as baths combined with using pour-on or sheep fat (a homemade cream made from sheep fat extraction). However, skin problems re-emerged when the animals returned to their previous environments.

Upon clinical examination, the affected donkeys exhibited restlessness as demonstrated by the constant pawing of their limbs and frequent swaying of their ears and tails. Intense itching was evident from the scratching of various body parts against structures such as the wooden corral, fence posts, walls, rubbing of the head between limbs, and biting the injured limb areas. During containment of the donkeys for clinical examination, insect swarms surrounded the animals, causing unrest.

The injuries were primarily located on the head, depigmented areas of the muzzle and cheek (Figure 1), flanks, forelimbs, hind limbs, and scrotal sac (Figure 2). They are characterized by areas of alopecia with crusts on the epidermis that exhibit serosanguinous exudation. The skin appeared wrinkled, thickened, and swollen.

### 3.2. Laboratory Assessment of Arthropods

In total, 378 dipteran species belonging to the genus *Culicoides* were collected. The most abundant species was *Culicoides ocumarensis* Ortiz (229 specimens), followed by *Culicoides foxi* (66 specimens), *Culicoides insignis* (57 specimens), *Culicoides* sp1 (35 specimens), *Culicoides* sp2 (32 specimens), *Culicoides leopoldoi, Culicoides vernoni*, and *Culicoides* sp3 (1 specimen each).

### 3.3. Laboratory Tests Related to Donkeys

To differentiate it from other diseases affecting horse skin, a group of animals underwent ectoparasite research through skin scraping, along with hematological and biochemical tests. Skin scraping results were negative for ectoparasites.

Statistical analysis of the hemogram and serum biochemical data revealed that the total count of leukocytes, segmented neutrophils, and lymphocytes, as well as creatinine levels, in animals with allergic dermatitis were significantly higher than those in healthy animals. However, the eosinophil count in animals with allergic dermatitis was significantly lower than that in healthy animals (Table 2 and Table 3).

The histological changes in donkeys 3, 4, and 5 were characterized by moderate superficial dermatitis with irregular acanthosis of the epidermis and pronounced diffuse orthokeratotic hyperkeratosis. Furthermore, multifocal areas with crusts, moderate infiltration of intact and degenerated neutrophils (intraepidermal pustules), and moderate vacuolar degeneration of epidermal keratinocytes were observed. Moderate multifocal inflammatory infiltration of lymphocytes, plasma cells, and occasional neutrophils and eosinophils were observed in the superficial dermis (Figure 3). Areas with moderate multifocal degeneration of collagen fibers and a few macrophages with intracytoplasmic granular black pigment (pigment incontinence) were also observed. Toluidine blue, Gram, and Warthin–Starry staining did not reveal the presence of mast cells, Gram-negative and Gram-positive bacteria, or spirochetes. 

Additionally, an experimental treatment was administered to two other animals (donkeys 3 and 5). Donkey 3 was moved from the pasture to a 4 × 4 m^2^ stable on the farm, provided with food and water in suitable troughs, and topically treated with 10 mL TOPLINE^®^ Pour-on RED thrice (Fipronil at 1%) at intervals of 7 d. Donkey 5 was admitted to the Veterinary Hospital at the UFPA and kept in a stable, where it was topically treated with 150 mL of a mixture containing copaiba oil and Scott’s multivitamin emulsion (cod liver oil, 0.882 g; vitamin A, retinyl palmitate, 3795 IU; vitamin D, cholecalciferol, 379 IU) at a 50% ratio for 30 d using a brush. The treatment was administered daily for the first 5 d and subsequently at intervals of 48 h. Additionally, a control animal (donkey 4) was used, which remained untreated on the farm and was observed daily for 30 d.

Donkey 5 was treated with copaiba oil during its stay at the Veterinary Hospital of UFPA. During this period, a remarkable improvement in the clinical condition was observed, leading to discharge from the hospital. Following discharge, the animals were continuously monitored on the farm, and 6 months after returning to the property, a significant increase in rainfall in the region combined with the presence of insects coincided with the reappearance of skin lesions on the donkey, characterizing a recurrence of the previously treated clinical condition.

Head lesions improved on donkey 3 after treatment with TOPLINE^®^ Pour-on RED. However, the lesions on the pelvic and thoracic limbs worsened, and new lesions were noted. Improvement in clinical symptoms such as reduced restlessness, limb kicking, and ear shaking were visible from the second day in donkey 5 after topical treatment with a mixture of copaiba oil and a multivitamin emulsion. With continued treatment, clear hair regrowth in the alopecic areas was observed, along with an improvement in inflammatory signs, such as reduced erythema, decreased sensitivity, and less local crust formation (Figure 4). 

Donkey 4, which remained untreated on the farm, showed no clinical improvement and continued to have the same skin lesions.

During a visit to the rural farm in Santo Antônio do Tauá (farm 4), 7 months after attending to the animals with dermatitis, it was observed that the animals had recovered, even without treatment for the lesions.

## 4. Discussion

The findings of this study showed that allergic dermatitis in donkeys is caused by bites from dipterans of the genus *Culicoides*. The diagnosis was based on epidemiological and clinicopathological data, treatment response, insect identification, and the exclusion of differential diagnoses of other skin diseases.

Allergic dermatitis has been diagnosed in Santa Inês and Texel sheep breeds across various municipalities in Pará, with varying prevalence [7,9]. According to Martins et al. [32], constant water accumulation in lakes, streams, and pastures promotes the multiplication of these insects. In the state of Pará, the highest rainfall occurs from December to May. Consequently, the highest occurrence of allergic dermatitis in donkeys during the rainy season was associated with a peak population number of *Culicoides*. The association and emergence of allergic dermatitis is supported by the fact that in September of the same year, the herd from the farm in Santo Antônio do Tauá recovered, even without treatment, likely due to a period of low rainfall in the region and a subsequent decrease in the *Culicoides* population.

The diagnosis likely only occurs in adult Pêga donkeys due to the predominance of this breed in the state of Pará after their acquisition from other Brazilian states for mule production through crossbreeding with horses. However, the horses living with donkeys did not exhibit any clinical signs of this disease. A possible explanation could be that the donkeys originated from the dry and hot regions of Africa, such as Egypt and Sudan, and adapted to the northeastern region of Brazil. Upon reaching the Amazon Biome, this species encounters hot and humid environments that cause environmental stress. This, combined with rampant mosquito proliferation, has led to the emergence of this condition.

The clinical signs such as restlessness, intense itching, and biting of skin lesions exhibited by the donkeys were similar to those found in donkeys in Israel [33]. Similar clinical findings have been reported in horses [11,34,35], mules [10], and sheep [7,9,11,36]. The injuries found in the distal region of the limbs were the most noticeable during clinical examination, as these wounds were caused by self-mutilation due to itching. 

The macroscopic lesions such as areas of alopecia, rough and thickened skin, edema, crust formation, and serosanguinous exudation observed in the present study are associated with the clinical and epidemiological findings, indicating that the disease is seasonal allergic dermatitis. This is similar to the condition described in sheep, horses, and mules in other regions of Brazil [10,11,37].

The total leukocyte, segmented neutrophil, and lymphocyte counts in the donkeys with allergic dermatitis lesions were significantly higher than those of the donkeys without lesions. According to Miranda et al. [38], these findings may be because of catecholamines linked to fear, excitement, and exercise, which shift leukocytes from the marginal blood compartment to the circulating compartment. The degree of neutrophilia can be up to twice the upper reference limit in horses [38]; however, this has not yet been reported in donkeys.

Immediate hypersensitivity reactions are characterized by the attraction of eosinophils to the inflammation site, thus increasing the perivascular infiltration of eosinophils, mast cells, and mononuclear cells into the dermis [6]. These factors may have contributed to lower circulating eosinophil counts in sick animals compared to in healthy animals.

Herein, although most biochemical analytes were within the normal ranges for Pêga breed donkeys [30,31], the total and direct bilirubin levels were below the reference values. This finding did not present clinical significance, and pre-analytical factors should be considered, as bilirubin is photosensitive and samples were collected approximately 1 h away from the laboratory where they were processed, which could introduce interference.

The histological lesions which were characterized by dermatitis were crucial in identifying the allergic cause, which is typical of type 1 hypersensitivity associated with the introduction of antigens through insect bites. These findings were similar to those reported by other authors who described this disease [9,36,37]. An allergic reaction can occur in response to proteins in insect saliva; thus, sensitive individuals produce IgE antibodies that specifically react with the allergen and trigger mast cell degranulation in the skin, which releases chemical mediators such as histamines [39,40]. The itching is triggered by the release of histamines from the granules of eosinophils, mast cells, and basophils, which are attracted by insect antigens. The presence of erosion and ulcer inflammatory infiltration observed in the lesions of the donkeys in the present case may be associated with self-trauma resulting from itching, as previously described in other studies [7,9,36,37]. 

Dipterans of the genus *Culicoides* cause allergic dermatitis in donkeys in Israel [33] and other domestic animal species such as sheep [9,33,37], mules [10], horses [41,42], and cattle [33].

*Culicoides* are cosmopolitan hematophagous insects [43]. According to Borkent and Dominiak [44], this genus includes approximately 1347 species worldwide. In the Neotropical region, at least 299 species are found, 151 of which are distributed throughout Brazil, with 123 species present in the Brazilian Amazon [45].

These insects primarily reproduce in wet or semi-aquatic environments, such as rivers, lakes, swamps, streams, tree holes, and animal dung environments, and were easily found on the studied farms throughout the year.

Donkey 5 was treated using a combination of copaiba oil and a multivitamin emulsion. This treatment protocol demonstrated satisfactory outcomes, characterized by significant clinical improvement in the animal over 30 d. According to Garcia and Yamaguchi [46] as well as Martins and Silva [47], copaiba oil restores membrane and mucosal functions, reduces edema and inflammation, modifies skin secretions, and promotes tissue healing. The anti-inflammatory, antimicrobial, analgesic, antiseptic, and healing activities of copaiba oil have been reported in other studies [48,49,50], but not in relation to allergic dermatitis in animals. 

The combination with a multivitamin emulsion enhanced the therapeutic effect of the oil due to its richness in vitamins A and D. Vitamin A plays a crucial role in epithelial tissue maintenance and development, collagen formation, and skin renewal, thereby improving skin elasticity, texture, and tone [51,52]. Vitamin D regulates keratinocyte and fibroblast proliferation and differentiation and contributes to collagen and cellular matrix maturation [53,54,55]. 

Using a pour-on on donkey 3 was limited to the head, possibly because the product was only distributed from the withers to the rump. Topical medication is necessary for all lesions associated with this type of disease, as pour-on application fails to repel insect attacks on the distal regions of the limbs. 

Although our study has limitations related to the small sample size for treatment, the authors suggest that these results demonstrate promising perspectives on the therapeutic potential of copaiba oil and a multivitamin emulsion for treating dermatitis. For future investigations, it is recommended to conduct more comprehensive studies and randomized clinical trials that consider the epidemiology and seasonal variations of the parasite, the natural course of the disease, the inclusion of a control group, and long-term observations of the groups subjected to the proposed treatments.

Differential diagnoses of other diseases that cause similar skin lesions, such as photosensitization, scabies, dermatophytosis, and dermatophilosis, should be performed. Although animals with lesions in depigmented areas of the skin were identified, it was possible to rule out a diagnosis of primary photosensitization, as there were no data indicating the presence of plants containing photosensitizing agents in the Amazon region. Furthermore, analyses of the clinical history and inspection of the properties did not reveal the existence of plants or substances that could induce photosensitization. The absence of characteristic lesions coupled with normal biochemical parameters is crucial for ruling out the differential diagnosis of secondary photosensitization caused by the ingestion of hepatotoxic plants, a condition previously reported in horses [56] and sheep [57] in the state of Pará. 

Scabies and dermatophytosis were ruled out because of the absence of mites and fungi in skin scrapings. Histochemical results also ruled out the potential involvement of coccoid/filamentous bacteria, intralesional bacteria, and spirochetes in the lesions associated with the skin disease. 

## 5. Conclusions

Epidemiological data, clinicopathological findings, and laboratory tests performed in conjunction with the exclusion of differential diagnoses were essential in diagnosing allergic dermatitis. Our study reports an association between allergic dermatitis and *Culicoides*. More comprehensive entomological studies are required to determine whether other dipteran species are also involved in the pathogenesis of allergic dermatitis in donkeys.

Although tested on only one donkey, topical treatment of allergic dermatitis caused by insect bites using the combined application of copaiba oil and a multivitamin emulsion demonstrated therapeutic potential, requiring further in-depth studies in the future.

The high-rainfall conditions in the Amazon Biome are conducive to the proliferation of *Culicoides* and the appearance of characteristic lesions of allergic dermatitis caused by insect bites.

## Figures and Tables

**Figure 1 animals-14-01330-f001:**
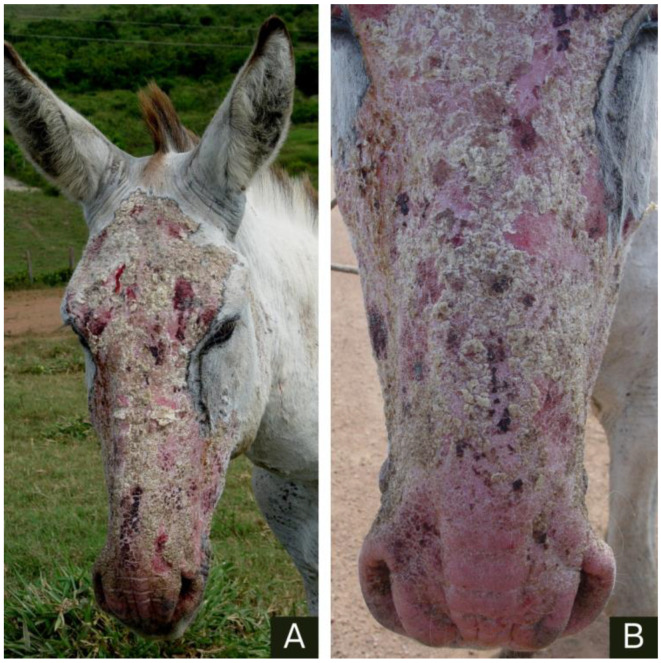
Donkey 1, an adult male Pêga breed, with allergic dermatitis due to *Culicoides* spp. bites. (**A**,**B**) Hyperemia and thickening of the skin in the facial region, on the lips, from the nostrils to the frontal sinus, and in the alopecic submandibular region.

**Figure 2 animals-14-01330-f002:**
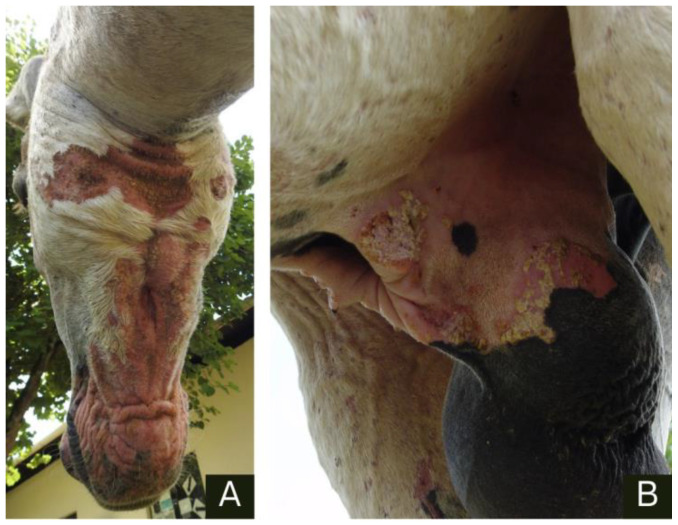
Donkey 5, an adult male Pêga breed, suffers from allergic dermatitis due to *Culicoides* spp. bites. (**A**) Alopecic, hyperemic, and thickened skin on the cheek. (**B**) Scrotal skin with hyperemia and crusts.

**Figure 3 animals-14-01330-f003:**
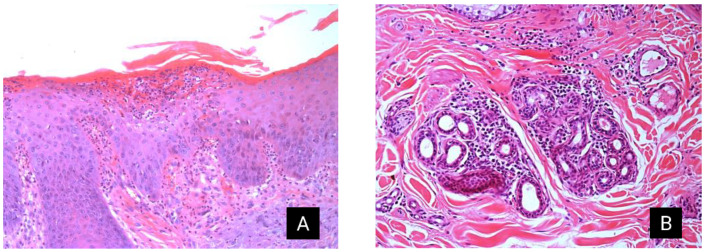
Allergic dermatitis in Pêga donkeys. (**A**) Haired skin, focal area of intraepidermal pustule filled with moderated intact and degenerated neutrophils, and swollen keratinocytes with vacuolated cytoplasm (edema). This pustule is covered by moderate keratin, cellular debris, and rare erythrocytes (crust). Multifocally, on the superficial dermis, there are moderate numbers of neutrophils, lymphocytes, plasma cells, and eosinophils. Furthermore, severe diffuse irregular acanthosis and moderate diffuse orthokeratotic hyperkeratosis are observed in the epidermis. Using H&E staining. (**B**) Haired skin: multifocally in the mid-dermis and surrounding adnexa, there is severe inflammatory infiltration of lymphocytes, plasma cells, neutrophils, and eosinophils. H&E, hematoxylin and eosin.

**Figure 4 animals-14-01330-f004:**
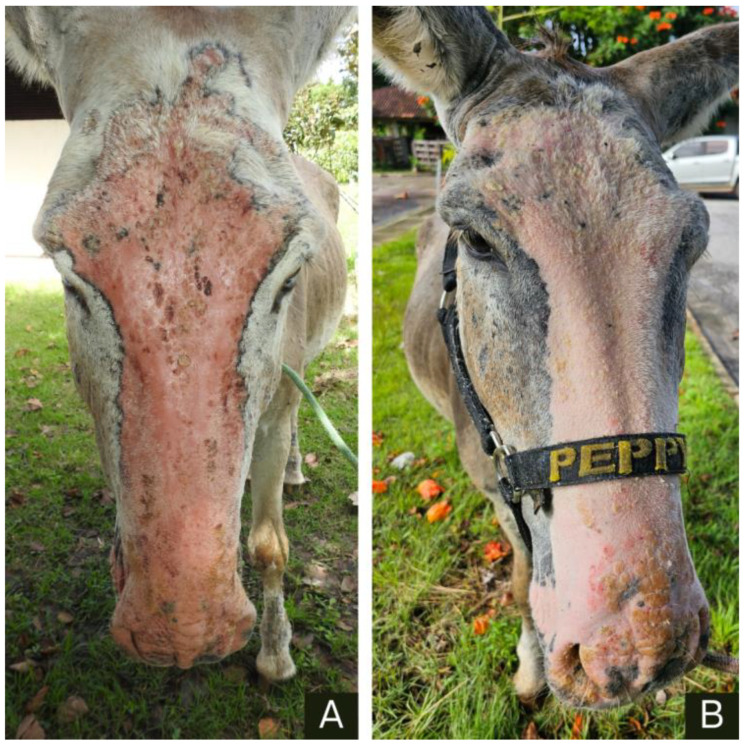
Donkey 5, an adult male Pêga breed, suffers from allergic dermatitis due to *Culicoides* spp. bites. (**A**) Alopecia and the presence of crusts on the skin from the nostrils to the hyperemic frontal sinus (pre-treatment). (**B**) Donkey 5, 30 d post-treatment with a mixture of copaiba oil and a multivitamin emulsion. Improvement in inflammation signs and hair regeneration is noted.

**Table 1 animals-14-01330-t001:** Incidence of morbidity, as an indication of allergic dermatitis, in donkeys in the northeast of Pará State, Amazon, Brazil.

Farm	Number of Donkeys on the Farm	Morbidity (%)
1	1	100 (1/1)
2	1	100 (1/1)
3	2	100 (2/2)
4	17	35.3 (6/17)
5	48	14.5 (7/48)

**Table 2 animals-14-01330-t002:** Mean, standard deviation, and *p*-value of hematological data from donkeys with and without allergic dermatitis lesions in the Amazon region, Pará, Brazil.

Hematological Data	Donkeys with Injury (n = 15)	Donkeys without Injury (n = 5)	*p*-Value
Red blood cells (×10^6^/µL)	6.3 ± 1.44	4.82 ± 1.39	0.0578
Hemoglobin (g/dL)	11.58 ± 0.98	11.72 ± 0.54	0.7671
Hematocrit (%)	34.80 ± 2.93	35.2 ± 1.64	0.7739
MCV (fL)	57.23 ± 10.73	64.2 ± 22.76	0.639
MCHC (%)	33.25 ± 0.07	33.26 ± 0.05	0.787
Total leukocytes (/µL)	16,148.47 ± 5475.48	15,050 ± 2909.68	0.00003 *
Monocytes (/µL)	430.73 ± 562.82	937 ± 1001.19	0.9686
Lymphocytes (/µL)	6181.80 ± 2374.42	5100.80 ± 76.82	0.00008 *
Eosinophils (/µL)	1764.27 ± 1308.86	1936.40 ± 701.34	0.008 *
Band neutrophil (/µL)	30.53 ± 65.14	145.40 ± 258.65	0.1129
Segmented neutrophil (/µL)	7741.33 ± 4110.06	6930.40 ± 1347.33	0.0008 *
Platelets (×10^3^/µL)	256.2 ± 162.28	208 ± 76.82	0.5416

* Significant statistical differences were determined using one-way analysis of variance, adopting a *p*-value = 0.05. n = number of animals. MCV, mean corpuscular volume. MCHC, mean corpuscular hemoglobin concentration.

**Table 3 animals-14-01330-t003:** Mean, standard deviation, and *p*-value of biochemical data from donkeys with and without allergic dermatitis lesions in the Amazon region, Pará, Brazil.

Biochemical Data	Donkeys with Injury (n = 15)	Donkeys without Injury (n = 5)	*p*-Value	References [30,31]
AST (U/L)	294.59 ± 86.71	317.74 ± 60.81	0.5958	262.15–436.63
GGT (U/L)	55.19 ± 16.56	51.22 ± 11.47	0.6328	38.05–86.17
Urea (mg/dL)	26.91 ± 9.21	24.3 ± 2.74	0.5522	24.39–38.03
Creatinine (mg/dL)	1.23 ± 0.30	0.9 ± 0.22	0.0327 *	0.93–1.37
Total bilirubin (mg/dL)	0.13 ± 0.06	0.17 ± 0.03	0.2015	0.16–0.48
Direct bilirubin (mg/dL)	0.04 ± 0.02	0.06 ± 0.02	0.2489	0.06–0.16
Indirect bilirubin (mg/dL)	0.09 ± 0.05	0.11 ± 0.03	0.2986	0.09–0.35

* Significant statistical differences were determined using one-way analysis of variance, adopting a *p*-value = 0.05. N = number of animals. AST, aspartate aminotransferase. GGT gamma-glutamyl transferase.

## Data Availability

The data were stored at the Federal University of Pará (UFPA) and are available upon request.

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
