# Peer review of "Allergic Dermatitis in Pêga Breed Donkeys (Equus asinus) Caused by Culicoides Bites in the Amazon Biome, Pará, Brazil"

_animals, 2024, doi:10.3390/ani14091330_

Round 1

Reviewer 1 Report

Comments and Suggestions for Authors

In the present manuscript, the authors describe a case series of allergic dermatitis in donkeys, caused by culicoides bites. The article does provide some important information about the epidemiology of this disease affecting donkeys from specific region of Brazil and some clinic-pathological aspects of the disease.

Despite describing some interesting information, a lot of vital=po details seems to be missing in the introduction. The introduction lacks important information about the pathogenesis of the lesions. Mosquito-bite dermatitis is triggered by a type I hypersensitivity, followed by a type IV hypersensitivity, and there is no mention of these important pathogenic steps in the text. This is crucial as the development of the disease is one of the main objectives of the project.

Also, other differentials for allergic dermatitis as well as the types of insects that causes allergic dermatitis in animals should be mentioned. Mosquito-borne diseases affecting equids (i.e., West Nile disease, Western and Eastern equine encephalomyelitis, etc.) should also be added as another possible consequence of mosquito bites.

The part of the study related with treatment should be removed. There is not enough data to correlate treatment success and clinical disease, since only 2 animals were evaluated. This part seems out of context in the manuscript and does not add value to the work.  The results seem somewhat empiric and does not prove causation.

Regarding the hematological and biochemical evaluation, some important analytes seem to be missing. There is some discussion during the text about chronicity of the lesions, but no inflammatory makers were analyzed. Haptoglobin and fibrinogen are important acute phase proteins that could give some insight about lesion development. Also, total proteins, including albumin and globulins were not analyzed.  Evaluating alpha, beta and gamma globulins could help elucidating the course of the disease, since alpha globulins tend to elevate in acute cases, and beta and gamma globulins elevate in cases of dermatopathies. Comparing the serum protein profile of affected with healthy animals could bring important information.

Some more specific suggestions below:

Intro:

Line 56: What aspect are you referring? Culicoid dermatitis? Please clarify.

Material and Methods:

Lines 106 – 107. Correct sentence order: “with and without the anticoagulant ethylenediaminetetraacetic acid (EDTA) for complete blood count and biochemical analyses respectively”.

Results:

Line 147. There is no mention to what animal from which farm had been suffering from lesion for over a year. Is this an average? Do you mean that all of them reported signs from over a year? More info and clarification are needed in this section.

Line 152. Please clarify if you mean all animals from all farms.

Lin 174. Serosanguineous exudation is not a sign of chronicity since it can be seen in the exudative stage of an acute inflammation.

Line 213. This sentence is incorrect. These special stains do not produce a positive or negative result. Each one of them is used to evaluate certain aspects of the tissue and results should be based on individual characteristics of the special stains.  For example, toluidine blue stains the granules of mast cells, so it would not be reported as negative, and rather with presence or absence of mast cells. Gram in the presence or absence of gram positive or negative bacteria and Warthin-starry for spirochetes (although it does stain all bacteria in general and is a good stain for bacterial morphology).

Figures

In general, the H&E stain is of poor quality. There is no contrast of the hematoxylin with the eosin. In 3B and 4B you can’t barely differentiate the cells on the inflammatory infiltrate.  Also, figure 4 does not add any useful information. An image showing the intracorneal pustules, that is hardly observed in figure 3A, should be added. Please add the tissue (Haired skin), the stain (e.g., H&E) and the magnificatioon the image description.

Discussion

Line 278. Again, serosanguinous exudation is not indicative of chronic disease.

Line 294. Reduction of water intake can indeed elevate creatinine levels due to prerenal azotemia, but urea should also be elevated. The authors are assuming without proper evidence that dehydration is causing elevation in creatinine, since no other parameter of dehydration were reported. There is no clinical evidence of dehydration, increase of serum albumin or evidence of hemoconcentration, since RBC, Hemoglobin and PCV are within normal range.

Line 302. When describing the histopathological changes, there is no mention of mast cells. They are important mediators of hypersensitivity. Toluidine blue was conducted but no results about presence of mast cells are included in the text.  There should be some discussion regarding this finding, as it is an indication of hypersensitivity.

Line 322. Treatment should be removed from the manuscript, as there is not enough information to prove causality.

Line 338. What about primary photosensitization? Was this excluded in these animals?

Line 346. Peas re -word as Dermatophilus are not spirochetes.

Line 347. The last sentence of the discussion seems out of context. What underscores the importance of additional tests?

Conclusions

Line 351. This sentence is not well structured. Do you mean the establishment of differential diagnoses?

Line 353. Please remove treatment.

Line 358. Replace “disease etiology” with pathogenesis of allergic dermatitis in donkeys.

Comments on the Quality of English Language

Please refer to main comments for specific suggestions regarding the English language

Author Response

Dear Reviewer,

We express our sincere gratitude for your contributions that have enhanced our manuscript. All suggestions have been duly considered and/or justified. The corresponding responses to each comment have been outlined below. The authors are available to clarify any doubts, address inquiries, or consider any additional suggestions that may arise.

Question: “The introduction lacks important information about the pathogenesis of the lesions. Mosquito-bite dermatitis is triggered by a type I hypersensitivity, followed by a type IV hypersensitivity, and there is no mention of these important pathogenic steps in the text.”

Answer: Suggestion accepted. Additional information incorporated into the introduction.

Question: “Also, other differentials for allergic dermatitis as well as the types of insects that causes allergic dermatitis in animals should be mentioned. Mosquito-borne diseases affecting equids (i.e., West Nile disease, Western and Eastern equine encephalomyelitis, etc.) should also be added as another possible consequence of mosquito bites.”

Answer: Suggestion accepted. Additional information incorporated into the introduction.

Question: “The part of the study related with treatment should be removed. There is not enough data to correlate treatment success and clinical disease, since only 2 animals were evaluated. This part seems out of context in the manuscript and does not add value to the work.  The results seem somewhat empiric and does not prove causation.”

Answer: We agree and acknowledge that the number of animals used to conclude such treatment is inadequate. However, the authors deem it important to retain the information related to the treatment, albeit in a more concise manner. The text has been adapted to emphasize the experimental nature, and additionally, the section regarding treatment has been moved from the methodology to the results as a supplementary point. The intention in preserving this information is to encourage future investigations with a more detailed methodology on this topic, especially considering that the efficacy of copaiba has already been validated in other studies. Therefore, the authors would like to keep the mentioned information about the treatment. However, if this poses an obstacle for the reviewer's ongoing evaluation, we kindly request that you inform us. We are available for any further clarifications.

Question:“Regarding the hematological and biochemical evaluation, some important analytes seem to be missing. There is some discussion during the text about chronicity of the lesions, but no inflammatory makers were analyzed. Haptoglobin and fibrinogen are important acute phase proteins that could give some insight about lesion development. Also, total proteins, including albumin and globulins were not analyzed.  Evaluating alpha, beta and gamma globulins could help elucidating the course of the disease, since alpha globulins tend to elevate in acute cases, and beta and gamma globulins elevate in cases of dermatopathies. Comparing the serum protein profile of affected with healthy animals could bring important information.”

Answer: The authors acknowledge and agree with the observation regarding the gap in our work, related to the absence of protein profile evaluation, an analysis that would have significant relevance to the study's structure. However, due to financial constraints faced in the Amazon region, we regrettably limited ourselves to the analysis of the most common analytes in clinical practice and available at the time of the study.

Question: Intro: Line 56: What aspect are you referring? Culicoid dermatitis? Please clarify.

Answer: We are referring to the limited study of diseases in donkeys in Brazil, despite their importance. The text has been modified for better comprehension. Suggestion accepted.

Question: Material and Methods: Lines 106 – 107. Correct sentence order: “with and without the anticoagulant ethylenediaminetetraacetic acid (EDTA) for complete blood count and biochemical analyses respectively”.

Answer: Suggestion accepted. The text has been modified for better comprehension.

Question: Results: Line 147. There is no mention to what animal from which farm had been suffering from lesion for over a year. Is this an average? Do you mean that all of them reported signs from over a year? More info and clarification are needed in this section.

Answer: This characteristic was consistently identified in all animals. Individuals younger than one year exhibited lesions for at least six months, while adult animals showed persistent lesions for over a year. This observation was noted on all properties. The suggestion was considered, and the text was adapted to promote better understanding.

Question: Line 152. Please clarify if you mean all animals from all farms.

Answer: Suggestion accepted. The text has been modified for better comprehension. 

Question: Lin 174. Serosanguineous exudation is not a sign of chronicity since it can be seen in the exudative stage of an acute inflammation.

Answer: Suggestion accepted. The text has been corrected.

Question: Line 213. This sentence is incorrect. These special stains do not produce a positive or negative result. Each one of them is used to evaluate certain aspects of the tissue and results should be based on individual characteristics of the special stains.  For example, toluidine blue stains the granules of mast cells, so it would not be reported as negative, and rather with presence or absence of mast cells. Gram in the presence or absence of gram positive or negative bacteria and Warthin-starry for spirochetes (although it does stain all bacteria in general and is a good stain for bacterial morphology).

Answer: Suggestion accepted. The text has been corrected.

Question: Figures: In general, the H&E stain is of poor quality. There is no contrast of the hematoxylin with the eosin. In 3B and 4B you can’t barely differentiate the cells on the inflammatory infiltrate.  Also, figure 4 does not add any useful information. An image showing the intracorneal pustules, that is hardly observed in figure 3A, should be added. Please add the tissue (Haired skin), the stain (e.g., H&E) and the magnification on the image description.

Answer: The authors appreciate the considerations made by the reviewer. Consequently, new slides have been remade, Figure 4 has been removed, and Figure 3 has been replaced with another of higher quality, as requested.

Question: Discussion: Line 278. Again, serosanguinous exudation is not indicative of chronic disease.

Answer: Suggestion accepted. The text has been corrected.

Question: Line 294. Reduction of water intake can indeed elevate creatinine levels due to prerenal azotemia, but urea should also be elevated. The authors are assuming without proper evidence that dehydration is causing elevation in creatinine, since no other parameters of dehydration were reported. There is no clinical evidence of dehydration, increase of serum albumin or evidence of hemoconcentration, since RBC, Hemoglobin and PCV are within normal range.

Answer: At the request of one of the reviewers, reference parameters have been added to the table of biochemical results. Upon a more detailed analysis of the literature, we opted to adopt the parameters described by Santos et al. (2018) due to their currency compared to the previously used reference. For the evaluation of bilirubin, we maintained the previous reference, as the most recent article does not mention it. As a result, the biochemical values were within the normal range, the discussion of which is included in the article.

Question: Line 302. When describing the histopathological changes, there is no mention of mast cells. They are important mediators of hypersensitivity. Toluidine blue was conducted but no results about presence of mast cells are included in the text.  There should be some discussion regarding this finding, as it is an indication of hypersensitivity.

Answer: Suggestion accepted. New slides have been prepared; however, we were unable to identify the presence of mast cells. We believe that the skin sampling may not have been sufficient due to the nature of the biopsy. The text has been corrected for better understanding.

Question: Line 322. Treatment should be removed from the manuscript, as there is not enough information to prove causality.

Answer: As previously emphasized, considering the potential of the information for future research, the authors would like to retain the treatment in the manuscript. However, the text has been modified and adapted to clarify the experimental nature of the treatment. However, if this poses an obstacle for the reviewer's ongoing evaluation, we kindly request that you inform us. We are available for any further clarifications.

Question: Line 338. What about primary photosensitization? Was this excluded in these animals?

Answer: The hypothesis of primary photosensitization has been refuted. Despite the observation of lesions in depigmented areas in some animals, the diagnosis of primary photosensitization was ruled out by epidemiological data, as the plants containing the active principle leading to direct photosensitization were not present. Additionally, the ingestion of photosensitizing plants or medications was not reported in history. The absence of hepatic alterations also supported the exclusion of this diagnosis.

Question: Line 346. Peas re -word as Dermatophilus are not spirochetes.

Answer: Suggestion accepted. The text has been corrected.

Question: Line 347. The last sentence of the discussion seems out of context. What underscores the importance of additional tests?

Answer: Suggestion accepted. The text has been modified for better comprehension.

Question: Conclusions: Line 351. This sentence is not well structured. Do you mean the establishment of differential diagnoses?

Answer: Suggestion accepted. The authors were referring to the exclusion of differential diagnoses. The text has been modified for better comprehension.

Question: Line 353. Please remove the treatment.

Answer: As previously emphasized, considering the potential of the information for future research, the authors would like to retain the treatment in the manuscript. However, the text has been modified and adapted to clarify the experimental nature of the treatment. However, if this poses an obstacle for the reviewer's ongoing evaluation, we kindly request that you inform us. We are available for any further clarifications.

Question: Line 358. Replace “disease etiology” with pathogenesis of allergic dermatitis in donkeys.

Answer: Suggestion accepted. The text has been corrected.

Reviewer 2 Report

Comments and Suggestions for Authors

Very well written manuscript, with logical continuity. I have nothing to fault the authors. For horse breeders, the results presented in the text are extremely valuable and useful for practice. However, I have one recommendation, that in section 2.5. Treatment - the authors describe what active substances are present in the TOPLINE preparation (line 129). It would also be useful to state specifically what vitamins are contained in the emulsion used (line 132).

I recommend to publish the manuscript.  

Author Response

Dear Reviewer,

We express our sincere gratitude for your contribution and recommendation to publish our manuscript. We have implemented your suggestions, and the text has been modified for better comprehension. The authors are available to clarify any doubts, address inquiries, or consider any further suggestions that may arise.

Question: Treatment - the authors describe what active substances are present in the TOPLINE preparation (line 129). It would also be useful to state specifically what vitamins are contained in the emulsion used (line 132).

Answer: Suggestion accepted. The text has been corrected.

Reviewer 3 Report

Comments and Suggestions for Authors

The authors describe the epidemiological, pathological and therapeutic aspects of allergic dermatitis in donkeys in Brazil as caused by Culicoides species bites. They study various aspects of the disease and describe clinical signs such as restlessness and intense itching. Skin lesions were observed across various body parts, featuring areas of alopecia with crusts and serosanguinous exudate. Their findings suggest a significant association between allergic dermatitis and Culicoides. The methodology was adequate and the results are described in detail and warrants publication. The manuscript is well written.

Some suggestions may be:

Culicoides are globally known vectors of a variety of livestock associated diseases. The authors can briefly refer to this and give an indication of the presence/abundance of these diseases in the study area.

Line 27: “insects” can be deleted.

Line 31: “identify the” can be given as “the identification of”.

Line 38: “(Diptera; Ceratopogonidae) can be inserted after specimens; Insert the authors name the first time a species is mentioned, i.e., Culicoides ocumarensis Ortiz. Amend throughout the manuscript.

Line 144: Insert grid references for the study area.

Line 150: “Incidence of allergic dermatitis in donkeys in the Northeast of Pará State, Amazon, Brazil.” The title of the Table can be given as: “Incidence of morbidity, as an indication of allergic dermatitis, in donkeys in the Northeast of Pará State, Amazon, Brazil”. The farms names (1 – 5) can be given as a subscript.

Line 152: “Urochloa humidicola” must be in italics.

Line 224: State how many light trap collections were made. Only 1? Since only the females are involved in blood feeding the sex ratio of the collection can be mentioned if available. Were any freshly blood fed specimens collected?

Author Response

Dear Reviewer,

All suggestions have been acknowledged and/or justified, and responses to each comment are provided below. We sincerely appreciate your contributions to the improvement of our manuscript. The authors are available to clarify any doubts, address inquiries, or consider any further suggestions that may arise.

Question: Culicoides are globally known vectors of a variety of livestock associated diseases. The authors can briefly refer to this and give an indication of the presence/abundance of these diseases in the study area.

Answer: Suggestion accepted. The text has been corrected, and the information has been added to the introduction.

Question: Line 27: “insects” can be deleted.

Answer: Suggestion accepted. The text has been corrected.

Question: Line 31: “identify the” can be given as “the identification of”.

Answer: Suggestion accepted. The text has been corrected.

Question: Line 38: “(Diptera; Ceratopogonidae) can be inserted after specimens; Insert the authors name the first time a species is mentioned, i.e., Culicoides ocumarensis Ortiz. Amend throughout the manuscript.

Answer: Suggestion accepted. The text has been corrected, and the information has been added.

Question: Line 144: Insert grid references for the study area.

Answer: Dear reviewer, we didn't quite understand. Could you please explain what you meant?

Question: Line 150: “Incidence of allergic dermatitis in donkeys in the Northeast of Pará State, Amazon, Brazil.” The title of the Table can be given as: “Incidence of morbidity, as an indication of allergic dermatitis, in donkeys in the Northeast of Pará State, Amazon, Brazil”. The farms names (1 – 5) can be given as a subscript.

Answer: Suggestion accepted. The text has been corrected.

Question: Line 152: “Urochloa humidicola” must be in italics.

Answer: Suggestion accepted. The text has been corrected.

Question: Line 224: State how many light trap collections were made. Only 1? Since only the females are involved in blood feeding the sex ratio of the collection can be mentioned if available. Were any freshly blood fed specimens collected?

Answer: The collections were conducted only once on each property. More collections could have been made; however, considering the quantity of insects obtained, we believe they were sufficient for identification purposes.

Dear reviewer, your consideration is highly relevant. Indeed, only females engage in blood feeding; however, information regarding the proportion of males to females has not been made available to us thus far. We appreciate your valuable input, as it will allow us to be attentive to similar situations in future experiments. Engorged females, meaning females with fully fed digestive systems, were not indicated in the sample spreadsheet analyzed. It is worth noting the importance of this information for future studies. Once again, we thank you for your considerations!

Reviewer 4 Report

Comments and Suggestions for Authors

The authors of the paper entitled ‘’Allergic Dermatitis in Pêga Breed Donkeys (Equus asinus) caused by Culicoides Bites in the Amazon Biome, Pará, Brazil’’ describe the epidemiological, clinical-pathological, and therapeutic aspects of allergic dermatitis in donkeys caused by Culicoides spp. bites. They found an association of allergic dermatitis in donkeys with Culidoides and approach a potential treatment of one allergic donkey using copaiba oil and multivitamin emulsion that shows promising results. 

Major comments:

·      Treatment with n=1 with short follow-up is probably too little to conclude on potential treatment

Minor comments: 

·      Figure 1 A and B: is there a sign of infection in the skin of the donkey? Also, how can photosensitization be excluded, please specify, especially since the lesion is on the depigmented part of the skin. Please comment. 

·      Line 189 and 285-286: No sign of neutrophils counts in Table 2: Hematological data. 

Please comment.

·     Line 191-192: However, the eosinophil count of the animals with allergic dermatitis was significantly lower than that of the healthy ones (Tables 2 and 3): No eosinophils   data in Table 3. Please correct. 

·     Table 3: Can the authors include the healthy range of those measurement in the table? 

·      Donkey 5 treatment: Is there other time points where the donkey continues to show improvement for the treatment: as 2 ,3 and 4 months post treatment? Please comment.  The treatment of donkey 5 can’t really tell if the recovery come from the copaiba oil itself or from the daily care of this donkey or just the natural course of disease. The control donkey 3 should be treated similarly but with another product (water per example). 

·      Line 267-269: A possible explanation that horses didn’t show any symptoms compared to donkeys is the origin of the donkey as assumed by the authors to be originally coming from a dry and hot region. What about the goats and sheep that showed similar skin symptoms?  Can you please also include a table with data regarding donkeys (breed, origin, and gender). Is there any comparison between allergic horses coming from dry and hot region to local ones? 

·      Line 356-358: Authors conclude that their strong association between allergic dermatitis and Culicoides.  However, in their results part, mainly descriptive and based on excluding other cause of symptoms is based on epidemiological data, clinical-pathological findings, and laboratory test specially the blood count. I would rather be careful with the word ‘’strong association’’. For a strong association, per example, authors could investigate PBMC response to culicoides from healthy and allergic donkeys blood by checking modulating immune response genes and cytokines for a strong association between allergic dermatitis and culicoides. Please replace “strong association”.

·      Histological sections in Figure 3 and 4: please add better quality with higher resolution

Author Response

Dear Reviewer,

All suggestions have been acknowledged and/or justified, and responses to each comment are provided below. We sincerely appreciate your contributions to the improvement of our manuscript. The authors are available to clarify any doubts, address inquiries, or consider any further suggestions that may arise.

Question: Treatment with n=1 with short follow-up is probably too little to conclude on potential treatment

Answer: We agree and recognize that the number of animals used to conclude a treatment of this nature is inadequate. However, the authors deem it important to retain the information related to the treatment, albeit in a more concise manner. The text has been adapted to emphasize the experimental nature, and additionally, the section regarding treatment has been moved from the methodology to the results as a supplementary point. The intention in preserving this information is to encourage future investigations with a more detailed methodology on this topic, especially considering that the efficacy of copaiba has already been validated in other studies. Therefore, the authors would like to keep the mentioned information about the treatment. However, if this poses an obstacle for the reviewer's ongoing evaluation, we kindly request that you inform us. We are available for any further clarifications.

Question: Figure 1 A and B: is there a sign of infection in the skin of the donkey? Also, how can photosensitization be excluded, please specify, especially since the lesion is on the depigmented part of the skin. Please comment.

Answer: The lesion exhibited characteristics more consistent with a hypersensitivity reaction, manifesting with a dry appearance. This finding was corroborated by the results of the histopathological analysis, which did not reveal evidence of bacterial infection. Regarding photosensitization, the text has been revised and supplemented with additional information for a better elucidation of the exclusion of this differential diagnosis.

Question: Line 189 and 285-286: No sign of neutrophils counts in Table 2: Hematological data. Please comment.

Answer: Suggestion accepted. The table has been corrected for better comprehension.

Question: Line 191-192: However, the eosinophil count of the animals with allergic dermatitis was significantly lower than that of the healthy ones (Tables 2 and 3): No eosinophils   data in Table 3. Please correct.

Answer:  Suggestion accepted. The text has been corrected.

Question: Table 3: Can the authors include the healthy range of those measurement in the table? 

Answer: After a more thorough analysis of the literature, we identified an article published in 2018 that covers a wide range of parameters we are investigating. Considering its relevance and inclusion of data regarding both males and females, which is pertinent to our study, we have chosen to use it as the primary source. Thus, no significant deviations were observed in relation to the established reference values. Regarding bilirubin, no clinically relevant results were observed; however, it is important to highlight the need to consider pre-analytical factors, as bilirubin is susceptible to photodegradation, and the analyzed samples were exposed for approximately 1 hour before being processed in the laboratory, which may have influenced the results. In this context, the text has been reviewed and updated as necessary.

Question: Donkey 5 treatment: Is there other time points where the donkey continues to show improvement for the treatment: as 2 ,3 and 4 months post treatment? Please comment.  The treatment of donkey 5 can’t really tell if the recovery come from the copaiba oil itself or from the daily care of this donkey or just the natural course of disease. The control donkey 3 should be treated similarly but with another product (water per example).

Answer: Animal 5 was admitted to the veterinary hospital, where it received treatment with copaiba oil, showed clinical improvement, was discharged, and subsequently followed up on the farm. After six months back on the property, with an increase in rainfall in the region associated with the presence of mosquitoes, the donkey experienced a recurrence of the lesions. Regarding the treatment, as previously emphasized, considering the potential of the information for future research, the authors would like to retain the treatment in the manuscript. However, the text has been modified and adapted to clarify the experimental nature of the treatment. However, if this poses an obstacle for the reviewer's ongoing evaluation, we kindly request that you inform us. We are available for any further clarifications.

Question: Line 267-269: A possible explanation that horses didn’t show any symptoms compared to donkeys is the origin of the donkey as assumed by the authors to be originally coming from a dry and hot region. What about the goats and sheep that showed similar skin symptoms?  Can you please also include a table with data regarding donkeys (breed, origin, and gender). Is there any comparison between allergic horses coming from dry and hot region to local ones? 

Answer: The researchers do not have this specific data available. One hypothesis raised is that, in addition to the climatic conditions of the Amazon region, certain characteristics of the properties may have contributed to aggravating the situation, such as the presence of stagnant water and the rainy season. It was observed that when transferred to other locations, the animals showed clinical improvement. The text has been modified for better comprehension.

Question: Line 356-358: Authors conclude that their strong association between allergic dermatitis and Culicoides.  However, in their results part, mainly descriptive and based on excluding other cause of symptoms is based on epidemiological data, clinical-pathological findings, and laboratory test specially the blood count. I would rather be careful with the word ‘’strong association’’. For a strong association, per example, authors could investigate PBMC response to culicoides from healthy and allergic donkeys blood by checking modulating immune response genes and cytokines for a strong association between allergic dermatitis and culicoides. Please replace “strong association”

Answer:  Suggestion accepted. The text has been corrected.

Question: Histological sections in Figure 3 and 4: please add better quality with higher resolution.

Answer: As requested by another reviewer, Figure 4 has also been removed. and Figure 3 has been replaced with another of higher quality, as requested.

Round 2

Reviewer 1 Report

Comments and Suggestions for Authors

Dear Authors,

Thank you for submitting the revised version of your manuscript and for addressing my suggestions. I can observe significant improvements in the manuscript, which is commendable.

I agree with your decision to retain the treatment component, and I find that the edits align more closely with the experimental nature of the study. Hopefully, further studies will be undertaken to thoroughly evaluate the use of Copaiba oil on equine cutaneous lesions.

It is unfortunate that funding remains a constant challenge in many places, particularly in developing countries like Brazil. However, despite financial constraints, the authors have managed to conduct important and meaningful research.

I do not have any specific comments on the manuscript content. However, I suggest that a thorough review of the English language be conducted to ensure clarity and coherence throughout

Comments on the Quality of English Language

Pleas review the English language.

Author Response

Dear Reviewer,

We would like to reiterate our gratitude for your valuable contributions to our manuscript. Your suggestions have been of utmost importance to us. We wish to emphasize that the entire English text of the manuscript has been thoroughly reviewed by a specialized company, and we trust that it is in compliance. However, we are available to make any necessary modifications if required.

Thank you for your continued attention and collaboration.